# Should Contrast-Enhanced Harmonic Endoscopic Ultrasound Be Incorporated into the International Consensus Guidelines to Determine the Appropriate Treatment of Intraductal Papillary Mucinous Neoplasm?

**DOI:** 10.3390/jcm10091818

**Published:** 2021-04-22

**Authors:** Tomohiro Yamazaki, Mamoru Takenaka, Shunsuke Omoto, Tomoe Yoshikawa, Rei Ishikawa, Ayana Okamoto, Atsushi Nakai, Kosuke Minaga, Ken Kamata, Kentaro Yamao, Yoriaki Komeda, Tomohiro Watanabe, Naoshi Nishida, Keiko Kamei, Ippei Matsumoto, Yoshifumi Takeyama, Takaaki Chikugo, Yasutaka Chiba, Masatoshi Kudo

**Affiliations:** 1Department of Gastroenterology and Hepatology, Faculty of Medicine, Kindai University, Osaka-Sayama 589-8511, Japan; chochiko.4kg@gmail.com (T.Y.); shunsuke.oomoto@gmail.com (S.O.); t.yoshikawa113@gmail.com (T.Y.); rei_i_0419@yahoo.co.jp (R.I.); a-o-k@mail.goo.ne.jp (A.O.); nakai_agmc@yahoo.co.jp (A.N.); kousukeminaga@yahoo.co.jp (K.M.); ky11@leto.eonet.ne.jp (K.K.); yamaken_volvo@yahoo.co.jp (K.Y.); y-komme@mvb.biglobe.ne.jp (Y.K.); tomohiro@med.kindai.ac.jp (T.W.); naoshi@med.kindai.ac.jp (N.N.); m-kudo@med.kindai.ac.jp (M.K.); 2Division of Hepato-Biliary-Pancreatic Surgery, Department of Surgery, Faculty of Medicine, Kindai University, Osaka-Sayama 589-8511, Japan; keiko-kamei@med.kindai.ac.jp (K.K.); ippeimm@gmail.com (I.M.); takeyama@med.kindai.ac.jp (Y.T.); 3Department of Pathology, Faculty of Medicine, Kindai University, Osaka-Sayama 589-8511, Japan; tchikugo@med.kindai.ac.jp; 4Clinical Research Center, Faculty of Medicine, Kindai University, Osaka-Sayama 589-8511, Japan; chibay@med.kindai.ac.jp

**Keywords:** contrast-enhanced harmonic endoscopic ultrasound, diagnostic value, international consensus guidelines, intraductal papillary mucinous neoplasm, pancreas

## Abstract

This study aimed to investigate whether the incorporation of contrast-enhanced harmonic endoscopic ultrasound (CH-EUS) into the international consensus guidelines (ICG) for the management of intraductal papillary mucinous neoplasm (IPMN) could improve its malignancy diagnostic value. In this single-center retrospective study, 109 patients diagnosed with IPMN who underwent preoperative CH-EUS between March 2010 and December 2018 were enrolled. We analyzed each malignancy diagnostic value (sensitivity (Se), specificity (Sp), positive predictive value (PPV), and negative predictive value (NPV)) by replacing fundamental B-mode EUS with CH-EUS as the recommended test for patients with worrisome features (WF) (the CH-EUS incorporation ICG) and comparing the results to those obtained using the 2017 ICG. The malignancy diagnostic values as per the 2017 ICG were 78.9%, 42.3%, 60.0%, and 64.7% for Se, Sp, PPV, and NPV, respectively. The CH-EUS incorporation ICG plan improved the malignancy diagnostic values (Se 78.9%/Sp, 53.8%/PPV, 65.2%/NPV 70.0%). CH-EUS may be useful in determining the appropriate treatment strategies for IPMN.

## 1. Introduction

Intraductal papillary mucinous neoplasms (IPMNs) are cystic lesions of the pancreas that are commonly encountered in daily practice. IPMN has been designated as an independent disease by the World Health Organization (WHO) [1]. IPMN includes a wide spectrum of lesions with varying degrees of histological differentiation from low-grade dysplasia (LGD) to high-grade dysplasia (HGD) and invasive carcinoma. Among these, HGD and invasive carcinoma are indicated for surgery because of their poor prognosis [2,3,4,5,6]. However, since it is very difficult to verify the pathological diagnosis before surgery, many patients who undergo surgical resection are overtreated. For the appropriate management of patients with IPMN, in 2006, the first edition of the international consensus guidelines (ICG) for the management of IPMN was published and revised in 2012 and subsequently in 2017 [6,7,8].

In recent years, the usefulness of EUS-FNA for malignant evaluation of IPMN has been reported [9]. Recent studies have also reported that a more accurate diagnosis of pancreatic cystic lesions is possible by combining various molecular markers [10]. However, on the other hand, there are also reports of dissemination of EUS-FNA for cystic diseases, and the indication of EUS-FNA for IPMN is still controversial [11].

In the 2017 ICG, the diagnosis of the mural nodule (MN) plays an important role to judge whether HRS or WF in the current ICG for the appropriate management of patients with branch duct-type (BD-type) IPMN. The “enhancing” MN assessed by contrast-enhanced CT (CE-CT) is one of the “high-risk stigmata (HRS)” of malignancy findings. The definite MN assessed by fundamental B-mode endoscopic ultrasound (FB-EUS) or positional transformation is the malignant finding within the IPMNs with “worrisome features (WF)”. Surgery is recommended in cases where these signs are present. In other words, to further develop the 2017 ICG for a more appropriate evaluation of surgical indications, a more diagnostic MN blood flow presence assessment modality is required.

The contrast-enhanced harmonic EUS (CH-EUS) has been developed as a novel imaging modality that can visualize the blood flow in the fine vessels using an ultrasonographic contrast agent. CH-EUS has been reported to be useful in the differential diagnosis of solid pancreatic tumors [12,13,14,15,16] and evaluation of malignancy of IPMN [16,17,18,19,20]. Figure 1 shows a case in which CH-EUS was useful for nodule evaluation; FB-EUS showed findings suspicious of nodules, and CH-EUS was used to diagnose the nodules. Pathological examination revealed this tumor as HGD (Figure 1).

Figure 2 shows a case in which CH-EUS was useful in differentiating a mucous mass; FB-EUS showed findings suspicious of a nodule, but CH-EUS showed no contrast effect, leading to the diagnosis of a mucous mass. Pathological examination revealed this tumor as LGD (Figure 2).

The ability of CH-EUS to identify MNs in IPMN has been reported to be superior to that of FB-EUS with positional transformation and color Doppler examination [21,22,23,24,25]. However, CH-EUS was incorporated into the current 2017 ICG.

We hypothesized that the incorporation of CH-EUS into the 2017 ICG would enable an improved diagnosis of the malignancy of BD-type IPMN. Improved preoperative diagnosis of the malignancy of IPMNs would reduce the number of surgical cases of benign IPMNs. This is the first study aimed to evaluate the utility of incorporating CH-EUS into the 2017 ICG for the management of IPMN.

## 2. Materials and Methods

### 2.1. Patients

We retrospectively analyzed the data obtained from 138 consecutive patients with a histopathological diagnosis of IPMN by surgical resection between March 2010 and December 2018 at Kindai University Hospital. The medical records of these patients were examined, and data regarding the patients’ characteristics, including age, sex, morphological features of IPMN (size, diameter of MPD, and nodule height), morphological classification of IPMN (main pancreatic duct (MD)-type, mixed-type, or BD-type), and pathological findings, were collected.

According to the 2017 ICG, MD-IPMN is defined as segmental or diffuse dilatation of the MPD of >5 mm without any dilatation of the branch duct or other causes of obstruction. BD-type IPMN is defined as a pancreatic cyst of >5 mm in diameter that communicates with the MPD. Mixed-type IPMN is defined as dilatation of the MPD of >5 mm with dilatation of the branch duct of >5 mm. The MPD diameter and cyst size were measured by magnetic resonance cholangiopancreatography (MRCP) in most cases.

Of the 138 cases, 109 patients were finally enrolled after excluding three patients who had concomitant pancreatic ductal adenocarcinoma, nine patients who had not undergone CE-CT or EUS, and 17 patients who were diagnosed with MD-type IPMN (Figure 3).

During the study period, the surgical indications were determined after discussions with surgeons using the 2012/2017 ICG as reference. In line with the Japanese guidelines, EUS-FNA was not performed [6,8,26,27]. This time there were 34 cases in which surgery was performed, although it was not indicated for surgery according to the 2012/2017 ICG. Of the 34 cases, 23 cases had a nodule height of 5 mm or less, 3 cases with false-positive pancreatic juice cytology, 3 cases with a cyst diameter showing an increasing tendency, and 5 cases with symptoms, such as bleeding or pancreatitis. Although these are not absolute surgical indications according to the 2012/2017 ICG, surgery was performed in consultation with patients and their families.

### 2.2. FB-EUS and CH-EUS

At our institution, when EUS is performed on IPMN cases, CH-EUS is actively performed in addition to FB-EUS in order to evaluate not only cysts but also the presence or absence of concomitant cancer.

An echoendoscope developed for conducting CH-EUS (GF-UCT260; Olympus Medical Systems, Tokyo, Japan) was used. EUS images were analyzed using the following imaging equipment: ALOKA Pro-Sound α10 (Hitachi, Tokyo, Japan) was used from March 2010 to March 2016 and ALOKA Pro-Sound F75 (Hitachi, Tokyo, Japan) was used from April 2016 to December 2018. After the evaluation of the pancreas and cysts using FB-EUS, the imaging mode was switched to the extended pure harmonic detection mode, which synthesized the filtered second-harmonic components with signals obtained from the phase shift for contrast-enhanced harmonic imaging. The transmitting frequency and mechanical indices were 4.7 MHz and 0.3, respectively. The ultrasound contrast agent used for CH-EUS was Sonazoid^®^ (Daiichi-Sankyo, Tokyo, Japan; GE Healthcare, Milwaukee, WI, USA), which consists of perfluorobutane microbubbles surrounded by a lipid membrane. Just before performing CH-EUS, the contrast agent was reconstituted with 2 mL of sterile water for injection, and a dose of 15 µL/kg body weight was prepared in a 2-mL syringe. A bolus injection of the ultrasound contrast agent was administered at a speed of 1 mL/s through a 22-gauge cannula placed in the antecubital vein, followed by a 10-mL saline solution flush to ensure that all contrast agents were introduced into the circulation. The CH-EUS examination was performed for approximately 60 s after the injection of a contrast medium. The presence or absence of blood flow in the nodule was evaluated for 20 s immediately after administration (vessel image), and the degree of contrast was evaluated during the period between 40 and 60 s after administration (perfusion image) (Figure 4).

All CH-EUS videos were stored and then individually reviewed by two EUS endoscopists with each of them having performed >1000 CH-EUS procedures. The kappa coefficient between the two endoscopists was 0.73. The final outcome of cases with different evaluations was decided in consultation.

### 2.3. Pathological Investigations

The specimens were serially transected at a thickness of 5–7 mm, and all slides were reviewed by a pathologist at the abovementioned institute. The pathological examination was carried out using hematoxylin and eosin staining and by recording the immunohistochemical reactivity against anti-mucin 1 (MUC1), MUC2, MUC5AC, and MUC6 antibodies in all cases according to the criteria defined by the Japanese Pancreas Society [28,29]. The tumors were classified as LGD, HGD, or invasive carcinoma. The T-staging was determined according to the 7th edition of the International Union against Cancer (UICC) Classification of Malignant Tumors [30,31].

### 2.4. Outcome Definitions

The primary aim of this study was to investigate whether the incorporation of CH-EUS into the 2017 ICG could improve the preoperative malignancy diagnostic value of IPMN. We assessed the impact of the CH-EUS incorporation into the 2017 ICG by replacing FB-EUS with CH-EUS as the recommended test for patients with WF on the diagnosis of malignancy (the “CH-EUS incorporation” ICG) (Figure 5 and Figure 6).

The malignant diagnostic ability between the 2017 ICG and the “CH-EUS incorporation” ICG were compared. The secondary aim was to examine the usefulness of CH-EUS for predicting malignant IPMN.

The study was approved by the Institutional Ethics Committee (IRB No. 31–144) and was conducted according to the provisions of the Declaration of Helsinki as revised in Fortaleza, Brazil in 2013.

### 2.5. Statistical Analysis

For the primary aim, among 109 enrolled cases of mixed or BD-type IPMNs, cases with histopathological diagnosis of HGD or invasive carcinoma were defined as malignant, and the malignancy diagnostic values in terms of the sensitivity (Se), specificity (Sp), positive predictive value (PPV), and negative predictive value (NPV) for each of the two strategies (2017 ICG/CH-EUS incorporation ICG) were analyzed and compared.

For the secondary aim, we performed a multivariable logistic regression analysis to explore factors predicting the diagnosis of IPMN, where the following factors were included as independent variables: CA 19-9, MPD diameter, cyst size, evaluation of the nodule height using CE-CT, evaluation of the nodule height using FB-EUS, and evaluation of the nodule height using CH-EUS as predictive factors of malignant IPMN; the results are shown by odds ratio (OR). Using a two-tailed test, *p* value < 0.05 were considered statistically significant. In addition, the optimal cut-off point and area under the curve (AUC) of the diagnostic performance of nodules evaluated by CH-EUS and FB-EUS in IPMN were calculated and compared through a receiver operating characteristic (ROC) analysis. The optimal cutoff point was calculated using (1-Se)^2^ + (1-Sp)^2^, defined as the minimum distance to the upper left corner.

All statistical analyses were performed using EZR (Saitama Medical Center, Jichi Medical University, Saitama, Japan), a graphical user interface for R (The R Foundation for Statistical Computing, Vienna, Austria). More precisely, it is a modified version of R commander designed to add statistical functions frequently used in biostatistics.

## 3. Results

### 3.1. Baseline Characteristics

Baseline characteristics of all 109 patients enrolled in this study are shown in Table 1. The patients’ mean age (range) was 69.5 (41–86) years, and 51.4% (56/109) were female. Regarding the location of the tumors, 65.1% (71/109) of the cases were located in the pancreatic head and 34.9% (38/109) was found in the body/tail of the pancreas. Macroscopically, 61 cases were classified as BD-type IPMN (56.0%, 61/109) and 48 cases as mixed-type IPMN (44.0%, 48/109). The histopathological results showed 52 cases (47.7%, 52/109) were LGD and 57 cases (52.3%, 57/109) were HGD or invasive carcinomas.

### 3.2. Investigation of Whether the Incorporation of CH-EUS into the 2017 ICG Could Improve the Malignancy Diagnostic Value of IPMN

Table 2 shows the comparison of the malignant diagnostic ability (Sensitivity/Specificity/PPV/NPV) between the 2017 ICG and “CH-EUS incorporation” ICG. With the 2017 ICG, the number of cases in which surgery was recommended (of which the number of cases pathologically diagnosed as malignant) was 75 (45), and 34 (12) were recommended for follow-up (Sensitivity 78.9% (45/57)/Specificity 42.3% (22/52)/PPV 60.0% (45/75)/NPV 64.7% (22/34)) (Figure 5).

On the other hand, with “CH-EUS incorporation” ICG, the number of cases in which surgery was recommended (of which the number of cases pathologically diagnosed as malignant) was 69 (45), and 40 (12) were recommended for follow-up (Sensitivity 78.9% (45/57)/Specificity 53.8% (28/52)/PPV 65.2% (45/69)/NPV 70.0% (28/40)) (Figure 6).

There was no significant difference between the 2017 ICG and “CH-EUS incorporation” ICG.

### 3.3. Examination of the Usefulness of CH-EUS for Predicting Malignant IPMN

The results of the multivariate analysis for the investigation of the predictors of malignant IPMN are shown in Table 3. In the current study, only the evaluation of the nodule height using CH-EUS was found to be an independent and significant prognostic factor for malignant IPMNs (OR 1.32; 95% CI 1.10–1.58; *p* = 0.002). Figure 7 shows a comparison of the ROC curves of nodule height evaluated using CH-EUS and FB-EUS. The highest diagnostic value of the evaluation of the nodule height using CH-EUS was calculated as 5 mm with an AUC of 0.764 (Sensitivity 55.8%, Specificity 84.2%) using the ROC curve analysis, whereas that of the evaluation of the nodule height using FB-EUS was calculated as 8 mm with an AUC of 0.714 (Sensitivity 67.3%, Specificity 64.9%). The AUC of CH-EUS was higher than that of FB-EUS (Figure 7).

## 4. Discussion

This is the first study that evaluated the utility of incorporating CH-EUS into the 2017 ICG for the management of IPMN. This study revealed that incorporating CH-EUS prevents mucus clots from being misdiagnosed as MN and improves the malignant diagnostic ability of the 2017 ICG. Changing the recommended test for patients with WF from FB-EUS to CH-EUS eliminated the possibility of mucus clots being misdiagnosed as MN. As a result, although the differences were not significant, improvements in Sensitivity (78.9% to 78.9%, *p* = 1.00), Specificity (42.3% to 53.8%, *p* = 0.33), PPV (60.0% to 65.2%, *p* = 0.61), and NPV (64.7% to 70.0%, *p* = 0.80) were observed.

In addition, the results of the multivariate and ROC curve analyses showed that the evaluation of contrast-enhanced MN >5 mm in diameter by CH-EUS had a high diagnostic value for malignant IPMNs.

When performing CH-EUS, the administered contrast medium first flows into the mass, and the contrast medium in the blood trembles due to ultrasonic waves to generate a harmonic echo in order to form an EUS image. If the lesion is a pancreatic tumor or a lymph node, both the determination of the presence or absence of blood flow and degree of contrast are required for the evaluation. Both “vessel image” and “perfusion image” are needed. However, there is a bias in the evaluation ability of the assessor.

On the other hand, if the lesion is an MN-like mass in IPMN, only the presence diagnosis is required. If the lesion is an MN with blood flow, the lesion is visualized by CH-EUS. If the lesion is a mucus clot, the lesion is visualized by FB-EUS, but not by CH-EUS because of the absence of blood flow. In this situation, only the “vessel image” is required, and a bias is unlikely to occur.

In this analysis, six patients who were diagnosed with definite MN by FB-EUS and underwent surgery were diagnosed with mucus clots without blood flow by CH-EUS. Six patients were diagnosed with mucus clots by CH-EUS and underwent surgery; two patients underwent surgery because the cyst diameter was >30 mm and they preferred to undergo surgery, and four patients underwent surgery because the cyst diameter tended to increase and they preferred to undergo the procedure. All these six cases were diagnosed with LGD by a pathological examination of surgical specimens; thus, if CH-EUS was incorporated into the ICG, resulting in the diagnosis of the nodule as a mucus clot, surgery could have been avoided. CH-EUS, which can easily distinguish MN from mucus clots, seems to match the ICG concept that anyone can easily use it.

In an examination of the malignancy predictors of IPMN, malignant nodules, dilated MPD, thickening of the cyst wall, and elevated CA 19-9 levels (>37 U/mL) were reported as malignancy predictors of IPMN [13]. However, the test that could diagnose malignant nodules has not been specified, and which test is used has a great influence on the evaluation of nodules. In the current study, the evaluation of the nodule height using CH-EUS was the only significant predictor of IPMN malignancy. Several studies have reported that the optimal cut-off value for the MN height is 5–10 mm [16,17,18,23,32]. Similar to the 2017 ICG, the European evidence-based guidelines for cystic pancreatic neoplasms also identified MN height of ≥5 mm as a predictor of IPMN malignancy [22]. In this study as well, the evaluation of the nodule height using CH-EUS on the ROC curve showed that the identification of nodules with a height of >5 mm was useful as a cut-off value for malignant tumors. On the other hand, in this analysis, the result of FB-EUS was a nodule height of >8 mm. It is thought that the fact that mucus attachment was also experienced in nodular lesions, and it is sometimes difficult to distinguish between MN and the surrounding cyst may be the cause of these differences.

It is thought that CH-EUS was able to measure the accurate nodular size in such cases, supporting the usefulness of CH-EUS.

### Why Is CH-EUS Not Adopted in ICG?

There are several reasons why CH-EUS has not been adopted in the 2017 ICG. The first reason is that the CH-EUS technique is not used globally yet, and for the guidelines to be recognized internationally, the guideline-formulating committee must be cautious regarding the inclusion of CH-EUS, regardless of its reported usefulness. Another problem is that the contrast agents used for CH-EUS differ from country to country, and there is a lack of standardization in this regard. Because the ultrasound settings are different for each contrast agent, the same effect may not be observed if the contrast agent is different from Sonazoid^®^ used in the present study. Moreover, the use of CH-EUS for the purpose of examination of pancreatic cysts is not covered by insurance companies in many countries including Japan.

However, as there is no doubt of the utility of CH-EUS in evaluating the treatment strategies for IPMN, the present analysis will be important in the future development and standardization of contrast agents and reforming the insurance norms in each country.

There are several limitations in this study. First, this was a retrospective single-center study with a small number of cases. Second, the results of this study were analyzed using only surgical cases. The target of the guideline is not only surgical cases but also all IPMN cases, including follow-up cases, and it is uncertain whether the results of the current study are applicable to all IPMNs. Moreover, the cases in this study were evaluated in a population that was not strictly indicated for surgery according to the 2017 ICG. The final pathological results showed that 47.7% of the 109 cases were benign. Since the study was conducted in a population with a large number of benign lesions, there is a bias as to whether the incorporation of CH-EUS into the 2017 ICG has the ability to accurately detect malignant IPMNs. Third, the contrast agent used in this study was Sonazoid^®^, which is not available worldwide. Therefore, the contribution of CH-EUS to the ICG presented in the current study may not be recognized in countries where Sonazoid^®^ is not available.

However, despite these limitations, we believe that the results of the current study will be useful in daily practice for the management of IPMN. CH-EUS, which can easily distinguish between MN and mucus clot, is expected to be useful for follow-up cases.

## 5. Conclusions

This study revealed that incorporating CH-EUS into the 2017 ICG would improve the diagnostic performance for malignant IPMNs. CH-EUS is likely to help refine the management strategies for patients with IPMN. CH-EUS includes an important potential benefit that can prevent unnecessary surgery. In the future, a multicenter prospective randomized comparison of FB-EUS and CH-EUS involving standardized surgical criteria and using the same contrast agent is desirable.

## Figures and Tables

**Figure 1 jcm-10-01818-f001:**
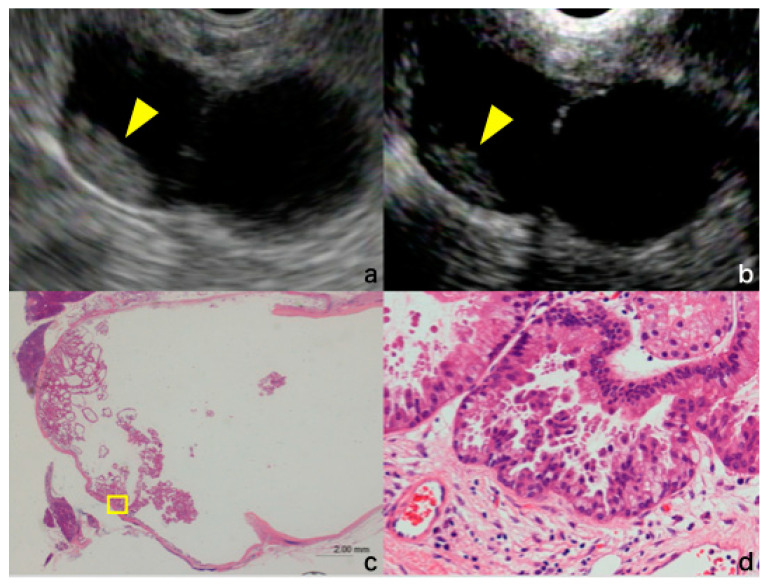
(**a**) Fundamental B-mode EUS showing a suspected mural nodule (arrowhead); (**b**) Contrast-enhanced harmonic EUS showing lesion enhancement and a definitive diagnosis of mural nodule (arrowhead); (**c**,**d**) Histopathological examination: structural atypia, nuclear enlargement, and irregular papillary structure are evident. EUS, endoscopic ultrasound.

**Figure 2 jcm-10-01818-f002:**
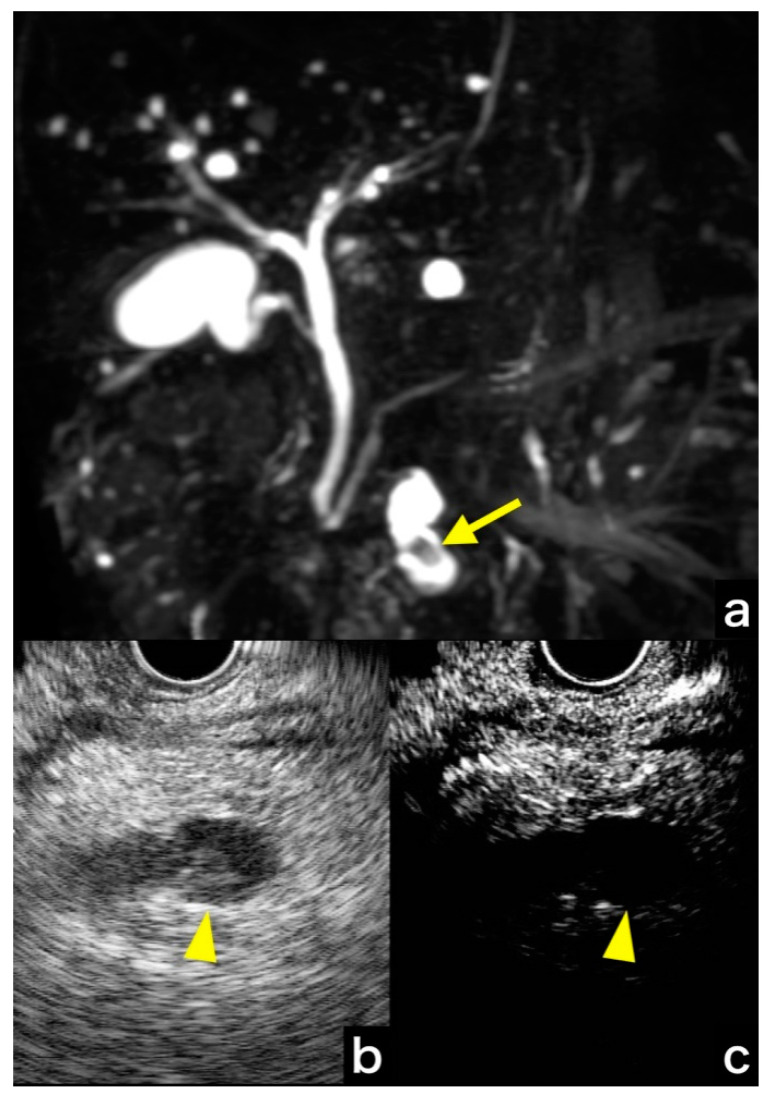
(**a**) MRCP shows a defect in the cystic lesion suspected as a mural nodule (arrow); (**b**) Fundamental B-mode EUS shows a suspected mural nodule (arrowhead); (**c**) Contrast-enhanced harmonic EUS shows the absence of enhancement (arrowhead). MRCP, magnetic resonance cholangiopancreatography; EUS, endoscopic ultrasound.

**Figure 3 jcm-10-01818-f003:**
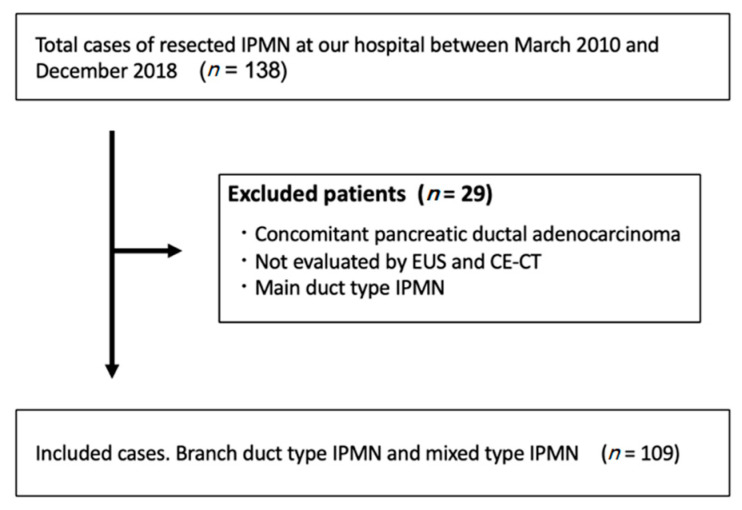
The flow chart outlining the enrollment of this study. Of the 138 cases who underwent surgical resection for IPMN, 109 patients were finally enrolled after excluding three patients who had concomitant pancreatic ductal adenocarcinoma, nine patients who had not undergone contrast-enhanced CT (CE-CT) or EUS, and 17 patients who were diagnosed with MD-type IPMN. EUS, endoscopic ultrasound; CT, computed tomography, IPMN, intraductal papillary mucinous neoplasm; MD, main pancreatic duct.

**Figure 4 jcm-10-01818-f004:**
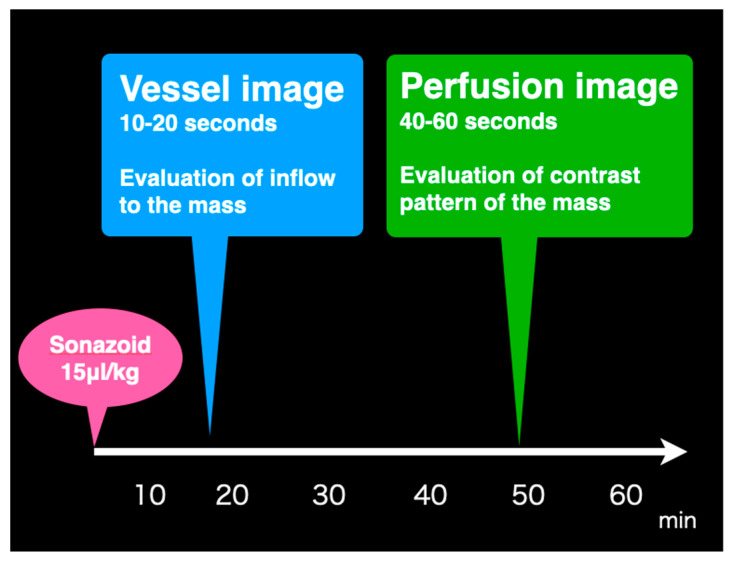
A schema rendering the time schedule of CH-EUS. The presence or absence of blood flow in the nodule was evaluated for 20 s immediately after administration (vessel image), and the degree of contrast was evaluated during the period between 40 and 60 s after administration (perfusion image). CH-EUS, contrast-enhanced harmonic endoscopic ultrasound.

**Figure 5 jcm-10-01818-f005:**
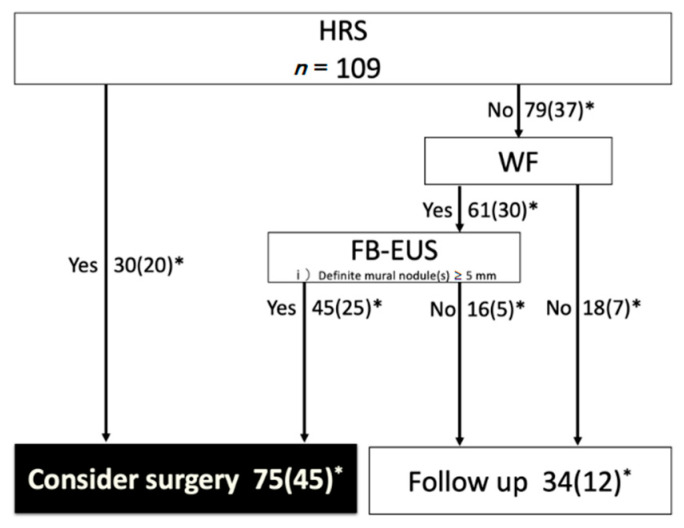
Algorithm for the management of BD-type IPMN as per the 2017 ICG. The recommended test for patients with WF on the diagnosis of malignancy is fundamental B-mode EUS. *, the number in parentheses is the number of malignant IPMNs; EUS, endoscopic ultrasound; WF, worrisome features; ICG, international consensus guidelines.

**Figure 6 jcm-10-01818-f006:**
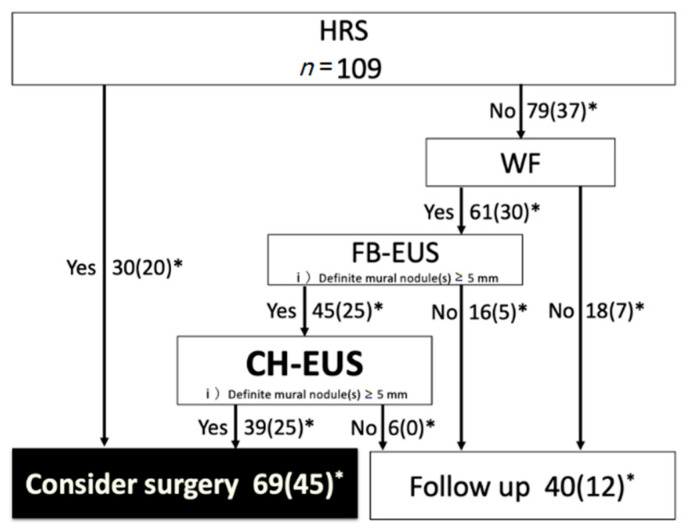
Algorithm for the management of BD-type IPMN as per the “CH-EUS incorporation” ICG. The recommended test for patients with WF on the diagnosis of malignancy was replaced FB-EUS with CH-EUS. *, The number in parentheses is the number of malignant IPMNs; EUS, endoscopic ultrasound; WF, worrisome features; ICG, international consensus guidelines; CH-EUS, contrast-enhanced harmonic endoscopic ultrasound.

**Figure 7 jcm-10-01818-f007:**
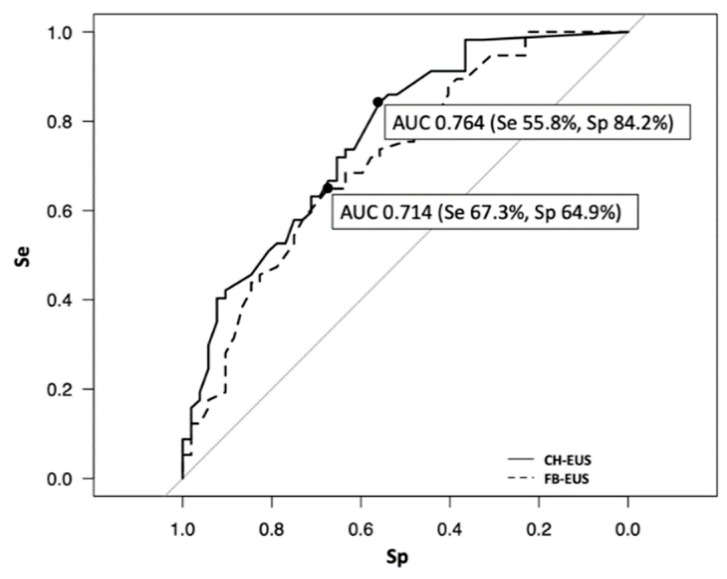
Comparison of the ROC curves of nodule height evaluated using CH-EUS and FB-EUS. The highest diagnostic value of the evaluation of the nodule height using CH-EUS was calculated as 5 mm with an AUC of 0.764 (Se 55.8%, Sp 84.2%) using the ROC curve analysis. The highest diagnostic value of the evaluation of the nodule height using FB-EUS was calculated as 8 mm with an AUC of 0.714 (Se 67.3%, Sp 64.9%) using the ROC curve analysis. CH-EUS, contrast-enhanced harmonic endoscopic ultrasound; FB-EUS, fundamental B-mode endoscopic ultrasound; ROC, receiver-operating characteristic; AUC, area under the curve.

**Table 1 jcm-10-01818-t001:** Baseline characteristics of the patients.

	Total (*n* = 109)
Sex (male:female), *n*	53:56
Age, mean (range), years	69.5 (41–86)
Morphology	
Cyst size, median (range), mm	28.1 (3–62)
Diameter of MPD, median (range), mm	5.7 (0.8–25)
Height of nodule *, median (range), mm	9.0 (1–52)
Cyst location, *n*	
Head	69
Body/tail	40
IPMN type (macro classification), *n*	
BD-type	61
Mixed-type	48
Pathology, *n*	
LGD	52
HGD	25
Invasive carcinoma	32

*, The height of nodule was assessed by contrast-enhanced harmonic endoscopic ultrasound. MPD; main pancreatic duct, BD-type; branched type, Mixed-type; mixed type, LGD; Low-grade dysplasia, HGD; High-grade dysplasia.

**Table 2 jcm-10-01818-t002:** A comparison of malignant diagnostic ability between 2017 ICG and CH-EUS incorporation ICG.

	2017 ICG	CH-EUS Incorporation ICG	*p*-Value
Sensitivty	78.9% (45/57)	78.9% (45/57)	1.00
Specificity	42.3% (22/52)	53.8% (28/52)	0.33
PPV	60.0% (45/75)	65.2% (45/69)	0.61
NPV	64.7% (22/34)	70.0% (28/40)	0.80

*p* < 0.05 was considered statistically significant. ICG; international consensus guidelines, CH-EUS; contrast-enhanced harmonic endoscopic ultrasound, PPV; positive predictive value, NPV; negative predictive value.

**Table 3 jcm-10-01818-t003:** The multivariate analysis for the examination of predictors of malignant IPMN.

Predictor	Univariate Analysis			Multivariate Analysis		
	OR	95%CI	*p*-Value	OR	95%CI	*p*-Value
CA19-9	1.01	0.99–1.02	0.46			
Total bilirubin	1.48	0.35–6.32	0.59			
Diameter of MPD	1.07	0.92–1.24	0.38	1.13	0.99–1.28	0.063
Size of IPMN	0.99	0.95–1.02	0.50			
Height of nodule measured by CE-CT	1.06	0.90–1.25	0.47			
Height of nodule measured by FB-EUS	0.90	0.78–1.04	0.15	0.91	0.80–1.02	0.12
Height of nodule measured by CH-EUS	1.30	1.08–1.55	<0.01	1.28	1.10–1.49	<0.01

*p* < 0.05 was considered statistically significant. OR; odds ratio, CI; confidence interval, MPD; main pancreatic duct, IPMN; intraductal papillary mucinous neoplasm, CE-CT; contrast enhanced computed tomography, FB-EUS; fundamental endoscopic ultrasonography, CH-EUS; contrast harmonic endoscopic ultrasonography.

## Data Availability

All the data used for this analysis can be confirmed at any time.

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
