# Peer review of "Should Contrast-Enhanced Harmonic Endoscopic Ultrasound Be Incorporated into the International Consensus Guidelines to Determine the Appropriate Treatment of Intraductal Papillary Mucinous Neoplasm?"

_jcm, 2021, doi:10.3390/jcm10091818_

Round 1

Reviewer 1 Report

Introduction: Authors to not highlight specifically what CH EUS will add over standard management. May be better if authors emphasize what are high risk stigmata and worrisome features in the ICG for IPMN. Then expand on how mural nodules are detected currently and the sens / spec of each technique. Then explain some of the differentials such as a "mucus ball". Explain how we try to differentiate currently (i.e. morphology / FNA) and some of the challenges particularly with FNA and trying to aspirate thick mucus from a small lesion and therefore the benefits of CH EUS.

Figure 1 and 2 - what type of IPMN were these ? malignant ? LGD

Results: Table 2 - the results look quite similar, were any significant differences observed?

How did CH EUS aid size assessment in MN? Why could this not be accurately measured with B-Mode EUS?

In malignant lesions that were hypoenhancing, did  CH EUS still measure the nodule accurately.

How did CH EUS perform in LGD, HGD and IC in IPMN?

Could authors differentiate between the MN and the surrounding cyst easily? Did this leadto over estimate the sizeofthe mural nodule in CH EUS.

In the 6 cases where CH EUS would have changed management, what did these patient have on final pathology ? LGD, or  HGD / IC?

Discussion: Additional questions which would benefit from discussion.

What sort of prospective study would be needed to prove utility of this technique would be needed before incorporation in to clinical algorithms?

Do these results apply to other contrast agents?

As all HRS and WF are clinical symptoms or morphological imaging findings it is difficult to see how the algorithm should change? Should more patients undergo EUS, which patients should undergo CH EUS?

What  are the time and cost implications for addition CH EUS to management algorithms in IPMN?

Author Response

Response to Reviewer 1

【Response】

Thank you very much for your very valuable remarks.

Your suggestions have improved our text very much.

We have made the following revisions to your suggestions.

Introduction: Authors to not highlight specifically what CH EUS will add over standard management. May be better if authors emphasize what are high risk stigmata and worrisome features in the ICG for IPMN. Then expand on how mural nodules are detected currently and the sens / spec of each technique. Then explain some of the differentials such as a "mucus ball". Explain how we try to differentiate currently (i.e. morphology / FNA) and some of the challenges particularly with FNA and trying to aspirate thick mucus from a small lesion and therefore the benefits of CH EUS.

【Response】

Thank you for your very important point.

First, we re-emphasized that the assessment of nodular lesions plays a very important role in the judgment of HRS and WF in the current ICG.

Also, thank you for pointing out that the explanation of the morphological evaluation of nodules and the histological evaluation by FNA was omitted.

We added the description of FNA to the introduction, and emphasized that we examined the usefulness of incorporating CH-EUS this time with the desire to further improve ICG.

In the 2017 ICG, the diagnosis of mural nodule (MN) plays an important role to judge whether HRS or WF in the current ICG for the appropriate management of patients with branch duct-type (BD-type) IPMN.

(Page 2 line 50-52)

In recent years, the usefulness of EUS-FNA for malignant evaluation of IPMN has been reported [9]. Recent studies have also reported that more accurate diagnosis of pancreatic cystic lesions is possible by combining various molecular markers [10]. but on the other hand, There are also reports of dissemination of EUS-FNA for cystic diseases, and the indication of EUS-FNA for IPMN is still controversial [11].

(Page 2 line 45-49)

Figure 1 and 2 - what type of IPMN were these? malignant ? LGD

【Response】

The target cases this time were all surgical cases, and pathological verification was performed. The case in Figure 1 was HGD and the case in Figure 2 was LGD.

We have modified the text as follows:

Pathological examination revealed this tumor as HGD (Page 2 line 64-65)

Pathological examination revealed this tumor as LGD (Page 2 line 73-74)

Results: Table 2 - the results look quite similar, were any significant differences observed?

【Response】

As your suggestion, we performed a statistically significant difference test and found no significant difference. Based on the result, Table2 has been recreated and the text has been added as follows.

There was no significant difference between the 2017 ICG and “CH-EUS incorporation” ICG. (Page 7 line 231-232)

Table 2. A comparison of malignant diagnostic ability between 2017 ICG and CH-EUS incorporation ICG.

2017 ICG

CH-EUS incorporation ICG

P-value

Sensitivty

78.9%

(45/57)

78.9%

(45/57)

1.00

Specificity

42.3%

(22/52)

53.8%

(28/52)

0.33

PPV

60.0%

(45/75)

65.2%

(45/69)

0.61

NPV

64.7%

(22/34)

70.0%

(28/40)

0.80

P < 0.05 was considered statistically significant. ICG; international consensus guidelines, CH-EUS; contrast-enhanced harmonic endoscopic ultrasound, PPV; positive predictive value, NPV; negative predictive value

Changing the recommended test for patients with WF from FB-EUS to CH-EUS eliminated the possibility of mucus clots being misdiagnosed as MN. As a result, although the differences were not significant, improvements in Sensitivity (78.9% to 78.9%, P=1.00), Specificity (42.3% to 53.8%, P=0.33), PPV (60.0% to 65.2%, P=0.61), and NPV (64.7% to 70.0%, P=0.80) were observed. (Page 9 line 267-271)

How did CH EUS aid size assessment in MN? Why could this not be accurately measured with B-Mode EUS?

【Response】

Thank you for valuable point out.

Since it is difficult to pathologically measure the size of IPMN nodular lesions, it is difficult to evaluate the validity of size measurement by pre-operative EUS.

In addition, mucus attachment has also been experienced with nodular lesions, making accurate nodular size difficult in such cases.

Also, as pointed out in the question below, it is sometimes difficult to distinguish between MN and the surrounding cyst in FB-EUS, and it tends to be overestimated the size of the mural nodule.

However, in CH-EUS, cysts are not contrasted and only MN is contrasted, so measurement of attached mucus is excluded, and the MN and the surrounding cyst can be distinguished.

CH-EUS is thought to be able to measure accurate nodule size.

We believe that one of the reasons why the cut-off point for malignant IPMN by ROC analysis was 8 mm in nodule height in FB-EUS but 5 mm in CH-EUS is due to the high nodule measurement ability of CH-EUS.

We have added the text as follows:

On the other hand, in this analysis, the result of FB-EUS was a nodule height of >8 mm.

It is thought that the fact that mucus attachment was also experienced in nodular lesions, and it is sometimes difficult to distinguish between MN and the surrounding cyst may be the cause of these differences.

It is thought that CH-EUS was able to measure accurate nodular size in such cases, supporting the usefulness of CH-EUS. (Page 9 line 307-312)

In malignant lesions that were hypo enhancing, did CH-EUS still measure the nodule accurately.

【Response】

As your point out, it is sometimes difficult to evaluate nodular lesions with very poor blood flow by CH-EUS. However, the judgment of nodule or mucus mass is not based on the contrast effect hyper or hypo, but on whether or not there is a contrast effect.

So, hypo enhancing lesions were evaluated as nodules this time.

How did CH EUS perform in LGD, HGD and IC in IPMN?

【Response】

Thank you for your comments. Since there is no pathology at the time of CH-EUS, we do not change the way we do CH-EUS according to the pathology.

Could authors differentiate between the MN and the surrounding cyst easily? Did this lead to overestimate the size of the mural nodule in CH-EUS.

【Response】

As mentioned above, in FB-EUS, it is sometimes difficult to differentiate between MN and the surrounding cyst, which tends to overestimate the size of the mural nodule. However, in CH-EUS, only the MN is contrasted, not the cyst, so it is thought to be possible to differentiate the MN from the surrounding cyst.

In the 6 cases where CH EUS would have changed management, what did these patient have on final pathology ? LGD, or  HGD / IC?

【Response】

All cases were LGD, as described below.

All these six cases were diagnosed with LGD by a pathological examination of surgical specimens(Page 9 line 291-292)

Discussion: Additional questions which would benefit from discussion.

What sort of prospective study would be needed to prove utility of this technique would be needed before incorporation into clinical algorithms?

【Response】

Thank you for your excellent points.

A multicenter prospective randomized comparison of FB-EUS and CH-EUS in IPMN scrutiny is needed to further prove the need to incorporate CH-EUS into clinical algorisms. However, it is quite difficult to do so for various reasons.

We have added the following into the last of conclusion part.

In the future, a multicenter prospective randomized comparison of FB-EUS and CH-EUS involving standardized surgical criteria and using the same contrast agent is desirable.

(Page 10 line 346-348)

Do these results apply to other contrast agents?

【Response】

If different contrast media are used, the resolution in EUS will also change, so the results of this study cannot be guaranteed.

This is described in the limitation part as below,

Third, the contrast agent used in this study was Sonazoid®, which is not available worldwide. Therefore, the contribution of CH-EUS to the ICG presented in the current study may not be recognized in countries where Sonazoid® is not available.

(Page 10 line 336-338)

As all HRS and WF are clinical symptoms or morphological imaging findings it is difficult to see how the algorithm should change? Should more patients undergo EUS, which patients should undergo CH EUS?

【Response】

We think your point is very important.

We believe that HRS decisions should be made conveniently in any hospital in any country. So, at this point, we think that it should be contrast-enhanced CT, not CH-EUS.

However, since the usefulness of CH-EUS was confirmed in this study as well, we believe that the spread of EUS examinations as well as contrast media will become important in the future, as described in the discussion part.

What are the time and cost implications for addition CH EUS to management algorithms in IPMN?

【Response】

The additional time by performing CH-EUS will be necessary to some extent. The issue of cost is very important and needs to be resolved urgently since there is no insurance approval for CH-EUS for pancreatic diseases in some countries. This point is also described in the discussion part as below,

Moreover, the use of CH-EUS for the purpose of examination of pancreatic cysts is not covered by insurance companies in many countries including Japan.

(Page 10 line 321-323)

Reviewer 2 Report

This is a very nice retrospective study evaluating the role of CH-EUS and how it could change management in the treatment of IPMNs. 

A few comments: 

  1. I am a little bit confused about some of the statements in the methods. In the paragraph that starts on line 172:
    1. The authors say that "34 cases were not indicated for 2017 IGG" - does this mean that those cases did not meet criteria for surgical resection based on the guidelines?
    2. The authors say that "EUS, which should not be performed unless the WF is positive, was performed". I don't quite understand this sentence, was EUS performed even though WF was not present? If so, why was EUS performed in those cases?
  2. In the results section, it would be helpful to specify in how many cases CH-EUS would have changed the management. While this is included in the change in sensitivity and specificity, I think it would be helpful to describe more explicitly. It is discussed in the discussion in lines 272, but it should also be in the results section.
  3. In the same paragraph as above (discussion line 272), the authors suggest that CH-EUS identified 6 patients with mucous clots but those patients went to surgery regardless for other indications (enlarging cyst size, patient preference). Was CH-EUS part of the decision process of whether patients were referred for surgical resection? This is not clear in the methods.
  4. It seems that CH-EUS was performed on all patients undergoing EUS for evaluation of IPMN. Was this part of a protocol? Or has CH-EUS been standard of care since 2010 at this center?
  5. I think preventing unnecessary surgery, especially a Whipple procedure which is a major surgical operation, is a very important potential benefit of this technology. I would emphasize this in the discussion/conclusion.

Author Response

Response to Reviewer 2

Comments and Suggestions for Authors

This is a very nice retrospective study evaluating the role of CH-EUS and how it could change management in the treatment of IPMNs.

【Response】

Thank you for your good evaluation.

A few comments:

I am a little bit confused about some of the statements in the methods. In the paragraph that starts on line 172:

1 The authors say that "34 cases were not indicated for 2017 IGG" - does this mean that those cases did not meet criteria for surgical resection based on the guidelines?

【Response】

I'm sorry to have confused you. As stated in the text, at our hospital the surgical indications were determined after discussions with surgeons using the 2012/2017 ICG as reference.

The details of 34 cases have been revised as follows in an easy-to-understand manner.

During the study period, the surgical indications were determined after discussions with surgeons using the 2012/2017 ICG as reference. In line with the Japanese guidelines, EUS-FNA was not performed [6,8,23,24]. This time there were 34 cases in which surgery was performed, although it was not indicated for surgery according to the 2012/2017 ICG. Of the 34 cases, 23 cases had a nodule height of 5 mm or less, 3 cases with false-positive pancreatic juice cytology, 3 cases with a cyst diameter showing an increasing tendency, and 5 cases with symptoms, such as bleeding or pancreatitis. Although these are not absolute surgical indications according to the 2012/2017 ICG, surgery was performed in consultation with patients and their families. (Page 4 line 113-121)

2 The authors say that "EUS, which should not be performed unless the WF is positive, was performed". I don't quite understand this sentence, was EUS performed even though WF was not present? If so, why was EUS performed in those cases?

【Response】

At our institution, when EUS is performed on IPMN cases, CH-EUS is actively performed in addition to FB-EUS in order to evaluate not only cysts but also the presence or absence of concomitant cancer.

We added this sentence into the “FB-EUS and CH-EUS”.

At our institution, when EUS is performed on IPMN cases, CH-EUS is actively performed in addition to FB-EUS in order to evaluate not only cysts but also the presence or absence of concomitant cancer. (Page 4 line 123-125)

In the results section, it would be helpful to specify in how many cases CH-EUS would have changed the management. While this is included in the change in sensitivity and specificity, I think it would be helpful to describe more explicitly. It is discussed in the discussion in lines 272, but it should also be in the results section.

【Response】

As your suggestion, we performed a statistically significant difference test and found no significant difference. Based on the result, Table2 has been recreated and the text has been added as follows.

There was no significant difference between the 2017 ICG and “CH-EUS incorporation” ICG. (Page 7 line 231-232)

Table 2. A comparison of malignant diagnostic ability between 2017 ICG and CH-EUS incorporation ICG.

2017 ICG

CH-EUS incorporation ICG

P-value

Sensitivty

78.9%

(45/57)

78.9%

(45/57)

1.00

Specificity

42.3%

(22/52)

53.8%

(28/52)

0.33

PPV

60.0%

(45/75)

65.2%

(45/69)

0.61

NPV

64.7%

(22/34)

70.0%

(28/40)

0.80

P < 0.05 was considered statistically significant. ICG; international consensus guidelines, CH-EUS; contrast-enhanced harmonic endoscopic ultrasound, PPV; positive predictive value, NPV; negative predictive value

Changing the recommended test for patients with WF from FB-EUS to CH-EUS eliminated the possibility of mucus clots being misdiagnosed as MN. As a result, although the differences were not significant, improvements in Sensitivity (78.9% to 78.9%, P=1.00), Specificity (42.3% to 53.8%, P=0.33), PPV (60.0% to 65.2%, P=0.61), and NPV (64.7% to 70.0%, P=0.80) were observed. (Page 9 line 267-271)

In the same paragraph as above (discussion line 272), the authors suggest that CH-EUS identified 6 patients with mucous clots but those patients went to surgery regardless for other indications (enlarging cyst size, patient preference). Was CH-EUS part of the decision process of whether patients were referred for surgical resection? This is not clear in the methods.

【Response】

This answer is the same as the answer to the question above, but I am sorry that the explanation of the surgical indications at our hospital is insufficient and confuses you. Regarding surgical indications, we have added the following.

Certainly, in these 6 cases, the nodule was diagnosed as a mucous mass by CH-EUS, so there was a possibility that surgery could be avoided. However, the surgery was performed for the reasons mentioned in the text.

During the study period, the surgical indications were determined after discussions with surgeons using the 2012/2017 ICG as reference. In line with the Japanese guidelines, EUS-FNA was not performed. (Page 4 line 113-115)

In this analysis, six patients who were diagnosed with definite MN by FB-EUS and underwent surgery were diagnosed with mucus clots without blood flow by CH-EUS. Six patients were diagnosed with mucus clots by CH-EUS and underwent surgery; two patients underwent surgery because the cyst diameter was >30 mm and they preferred to undergo surgery, and four patients underwent surgery because the cyst diameter tended to increase and they preferred to undergo the procedure. (Page 9 line 286-291)

It seems that CH-EUS was performed on all patients undergoing EUS for evaluation of IPMN. Was this part of a protocol? Or has CH-EUS been standard of care since 2010 at this center?

【Response】

Thank you for your question.

Same as the answer to the question above, but at our institution, when EUS is performed on IPMN cases, CH-EUS is actively performed in addition to FB-EUS in order to evaluate not only cysts but also the presence or absence of concomitant cancer. We have already introduced CH-EUS in 2010 and have conducted and reported on some clinical studies.

We added this sentence into the “FB-EUS and CH-EUS”.

At our institution, when EUS is performed on IPMN cases, CH-EUS is actively performed in addition to FB-EUS in order to evaluate not only cysts but also the presence or absence of concomitant cancer. (Page 4 line 123-125)

I think preventing unnecessary surgery, especially a Whipple procedure which is a major surgical operation, is a very important potential benefit of this technology. I would emphasize this in the discussion/conclusion.

【Response】

As your suggestion, we added the sentence “CH-EUS includes an important potential benefit that can prevent unnecessary surgery” into the conclusion part. (Page 10 line 345-346)

Reviewer 3 Report

This is a good article that helps to make the management guideline for IPMN more accurate using contrast-enhanced harmonic EUS (CH-EUS).
Although new guidelines for IPMN treatment continue to be introduced through the results of many studies, it is difficult to make decisions because there are many factors to consider in actual clinical practice. As introduced in this study, if CH-EUS is incorporated into IPMN treatment policy guidelines,  it will be useful in deciding the treatment strategy for IPMN.
However, I suggest some minor revisions.

1) In Table 2. I would like to indicate the statistical significance of the difference between the 2017 ICG and the CH-EUS incorporation ICG.

2) Correct 2017 IGG at line 108  --> 2017 ICG.

Author Response

Response to Reviewer 3

Comments and Suggestions for Authors

This is a good article that helps to make the management guideline for IPMN more accurate using contrast-enhanced harmonic EUS (CH-EUS).

Although new guidelines for IPMN treatment continue to be introduced through the results of many studies, it is difficult to make decisions because there are many factors to consider in actual clinical practice. As introduced in this study, if CH-EUS is incorporated into IPMN treatment policy guidelines, it will be useful in deciding the treatment strategy for IPMN.

However, I suggest some minor revisions.

【Response】

Thank you for good evaluation.

1) In Table 2. I would like to indicate the statistical significance of the difference between the 2017 ICG and the CH-EUS incorporation ICG.

【Response】

As your suggestion, we performed a statistically significant difference test and found no significant difference. Based on the result, Table2 has been recreated and the text has been added as follows.

There was no significant difference between the 2017 ICG and “CH-EUS incorporation” ICG. (Page 7 line 231-232)

Table 2. A comparison of malignant diagnostic ability between 2017 ICG and CH-EUS incorporation ICG.

2017 ICG

CH-EUS incorporation ICG

P-value

Sensitivty

78.9%

(45/57)

78.9%

(45/57)

1.00

Specificity

42.3%

(22/52)

53.8%

(28/52)

0.33

PPV

60.0%

(45/75)

65.2%

(45/69)

0.61

NPV

64.7%

(22/34)

70.0%

(28/40)

0.80

P < 0.05 was considered statistically significant. ICG; international consensus guidelines, CH-EUS; contrast-enhanced harmonic endoscopic ultrasound, PPV; positive predictive value, NPV; negative predictive value

Changing the recommended test for patients with WF from FB-EUS to CH-EUS eliminated the possibility of mucus clots being misdiagnosed as MN. As a result, although the differences were not significant, improvements in Sensitivity (78.9% to 78.9%, P=1.00), Specificity (42.3% to 53.8%, P=0.33), PPV (60.0% to 65.2%, P=0.61), and NPV (64.7% to 70.0%, P=0.80) were observed. (Page 9 line 267-271)

2) Correct 2017 IGG at line 108 --> 2017 ICG.

【Response】

We revised as your suggestions. The corrections are yellow highlighted.

Reviewer 4 Report

Thank you for your paper. I think it is of great interest and relevant to body of literature. 

Introduction

  1. 2nd sentence - please replace was with ' has been'.
  2. Line 43 - please add in the word 'subsequently in' after and but before 2017.

Methods

  1. section 2.2, line 147: please provide the kappa coefficient (k) between the two endoscopists please

Results

  1. line 202 - one decimal place please
  2. Table 2 - please write out in full sensitivity and specificity.

Discussion

  1. 1st paragraph - again please write the words sensitivity and specificity in full please.

Conclusion

  1. last sentence  - please rewrite as ' CH-EUS is likely to help refine the management strategies for patients with IPMN'.
  2. Also, may state that further prospective studies are required with the use of CH-EUS

Author Response

Response to Reviewer 4

Comments and Suggestions for Authors

Thank you for your paper. I think it is of great interest and relevant to body of literature.

【Response】

Thank you for good evaluation.

Introduction

  1. 2nd sentence - please replace was with ' has been'.
  2. Line 43 - please add in the word 'subsequently in' after and but before 2017.

【Response】

We revised as your suggestions. The corrections are highlighted in yellow.

IPMN has been designated as an independent disease by the World Health Organization (WHO). (Page 1 line 35-36)

For the appropriate management of patients with IPMN, in 2006, the first edition of the international consensus guidelines (ICG) for the management of IPMN was published and revised in 2012 and subsequently in 2017. (Page 1-2 line 41-44)

Methods

  1. section 2.2, line 147: please provide the kappa coefficient (k) between the two endoscopists please

【Response】

We provided the kappa coefficient (k) as your suggestions. The corrections are highlighted in yellow.

The kappa coefficient between the two endoscopists was 0.73. The final outcome of cases with different evaluations was decided in consultation. (Page 5 line 153-155)

Results

  1. line 202 - one decimal place please
  2. Table 2 - please write out in full sensitivity and specificity.

【Response】

We revised as your suggestions. The corrections are highlighted in yellow.

Discussion

  1. 1st paragraph - again please write the words sensitivity and specificity in full please.

【Response】

We revised as your suggestions. The corrections are highlighted in yellow.

Conclusion

  1. last sentence - please rewrite as ' CH-EUS is likely to help refine the management strategies for patients with IPMN'.
  2. Also, may state that further prospective studies are required with the use of CH-EUS

【Response】

We revised as your suggestions. The corrections are highlighted in yellow.

As your advice, we replace the sentence “In the future, a multicenter prospective randomized comparison of FB-EUS and CH-EUS involving standardized surgical criteria and using the same contrast agent is desirable.” from the last part of limitation to conclusion.

CH-EUS is likely to help refine the management strategies for patients with IPMN. CH-EUS includes an important potential benefit that can prevent unnecessary surgery. In the future, a multicenter prospective randomized comparison of FB-EUS and CH-EUS involving standardized surgical criteria and using the same contrast agent is desirable.

(Page 10 line 344-348)
